# *GBSOT4* Enhances the Resistance of *Gossypium barbadense* to *Fusarium oxysporum* f. sp. *vasinfectum* (FOV) by Regulating the Content of Flavonoid

**DOI:** 10.3390/plants12203529

**Published:** 2023-10-11

**Authors:** Zhanlian Su, Yang Jiao, Zhengwen Jiang, Pengfei Liu, Quanjia Chen, Yanying Qu, Xiaojuan Deng

**Affiliations:** 1College of Agriculture, Xinjiang Agricultural University, Urumqi 830052, China; 17799668420@163.com (Z.S.); jycotton@163.com (Y.J.); a1021240495@163.com (Z.J.); lpfei007@163.com (P.L.); xjyyq5322@126.com (Y.Q.); 2Cotton Research Institute, Xinjiang Academy of Agriculture and Reclamation Science, Shihezi 832000, China

**Keywords:** *Gossypium barbadense*, FOV, *SOT*, VIGS, flavonoid, WGCNA, metabolomics

## Abstract

Sulfotransferases (SOTs) (EC 2.8.2.-) are sulfate regulatory proteins in a variety of organisms that have been previously shown to be involved in regulating a variety of physiological and biological processes, such as growth, development, adaptation to land, stomatal closure, drought tolerance, and response to pathogen infection. However, there is a lack of comprehensive identification and systematic analysis of *SOT* in cotton, especially in *G. barbadense*. In this study, we used bioinformatics methods to analyze the structural characteristics, phylogenetic relationships, gene structure, expression patterns, evolutionary relationships, selection pressure and stress response of *SOT* gene family members in *G. barbadense*. In this study, a total of 241 *SOT* genes were identified in four cotton species, among which 74 *SOT* gene members were found in *G. barbadense*. According to the phylogenetic tree, 241 *SOT* protein sequences were divided into five distinct subfamilies. We also mapped the physical locations of these genes on chromosomes and visualized the structural information of *SOT* genes in *G. barbadense*. We also predicted the *cis*-acting elements of the *SOT* gene in *G. barbadense*, and we identified the repetitive types and collinearity analysis of *SOT* genes in four cotton species. We calculated the Ka/Ks ratio between homologous gene pairs to elucidate the selective pressure between *SOT* genes. Transcriptome data were used to explore the expression patterns of *SOT* genes, and then qRT-PCR was used to detect the expression patterns of *GBSOT4*, *GBSOT17* and *GBSOT33* under FOV stress. WGCNA (weighted gene co-expression network analysis) showed that *GB_A01G0479* (*GBSOT4*) belonged to the MEblue module, which may regulate the resistance mechanism of *G. barbadense* to FOV through plant hormones, signal transduction and glutathione metabolism. In addition, we conducted a VIGS (virus-induced gene silencing) experiment on *GBSOT4*, and the results showed that after FOV inoculation, the plants with a silenced target gene had more serious leaf wilting, drying and cracking than the control group, and the disease index of the plants with the silenced target gene was significantly higher than that of the control group. This suggests that *GBSOT4* may be involved in protecting the production of *G. barbadense* from FOV infection. Subsequent metabolomics analysis showed that some flavonoid metabolites, such as Eupatorin-5-methylether (3′-hydroxy-5,6,7,4′-tetramethoxyflavone, were accumulated in cotton plants in response to FOV infection.

## 1. Introduction

Sulfur is one of the most basic elements in plant life, and sufficient S in soil is conducive to the growth and development of plants and to obtaining a high quality and high yield of plants [1]. In mammals, sulfonation contributes to the homeostasis and regulation of many biologically active endogenous compounds [2], and in plants, sulfate coupling plays an important role in plant growth and development and stress adaptation, and the content of S element in any plant tissue is less than 0.25%, which may be considered to be seriously deficient in the S element [3].

Sulfotransferases (SOTs) (EC 2.8.2.-) are sulfate regulatory proteins in a variety of organisms. SOTs catalyze the transfer of sulfonate groups from 3′-phosphoadenosine 5′-phosphosulfate (PAPS) to appropriate hydroxyl/amino groups on many substrates; sulfate substances in plants function as secondary metabolites. Hormones are combined with stimulating situations, and important S-storage substances are used throughout the life cycle [4,5]. The first *SOT* gene was cloned from a flavonoid species (*Asteraceae*) and was related to the sulfation reaction of flavonols [6]. Subsequently, the encoded cDNA thiotransferase was isolated from *Arabidopsis thaliana*, and the derived 302 amino acid polypeptide was highly correlated with plant flavonol sulfotransferase [7]. In addition, SOT proteins are involved in regulating a variety of physiological and biological processes, such as growth, development, adaptation to land, stomatal closure, drought tolerance, and pathogen infection [8,9,10]. In addition, in order to further explore the biological function of *SOT* genes in plants, the *SOT* genes family in many species has been carefully studied, such as in *G. hirsutum* [11], *Solanum tuberosum* [12], *Brassica rapa* L. [13], *Rosaceae* species and three model plants [14].

Cotton is one of the most important fiber crops in the world. According to previous research reports, *G. barbadense* and *G. hirsutum* evolved directly through the hybridization and doubling of A and D genomes. *Gossypium arboreum* (A2 genome) is a donor of the A genome, and *G. raimondii* (D5 genome) is a donor of the D genome [15,16,17,18]. Cotton is often affected by abiotic stress factors such as drought, salt–alkali stress, cold, high temperature and disease during its growth and development. In the external environment of abiotic stress, it will affect the growth of cotton and reduce the yield and fiber quality of cotton. In China, Xinjiang is the main production area of island cotton, and wilt disease is the most important disease that affects the yield of island cotton [19]. Under normal circumstances, the survival time of this pathogen is five to ten years, and it begins to infect plants in the early stages of seed development, causing great impact on cotton production and fiber quality [20,21,22]. Therefore, improving plant stress resistance can enhance plant adaptability to a stress environment. With the completion of cotton genome sequencing, it provides a basis for the study of the *SOT* gene family.

In this study, based on previous QTL mapping results, major disease-resistance genes were screened, and the *GB_A01G0479* (*GBSOT4*) gene was identified as a candidate gene for resistance to Fusarium wilt in island cotton [23]. Subsequently, we conducted a comprehensive identification and characterization of *SOT* gene family members from two diploid cottons (*G. arboreum,* and *G. raimondii*) and two tetraploid cottons (*G. hirsutum*, *G. barbadense*). Phylogenetic analysis, gene structure, conserved motifs and sequence identification were used to identify the evolutionary relationships among *SOT* genes in cotton. Next, a collinearity analysis is performed for the non-synonymous (Ka) and synonymous (Ks) substitution ratio (Ka/Ks ratio). In addition, the expression of the *GBSOT* gene was analyzed by promoter *cis*-element analysis and tissue-specific expression pattern, and the function of the *GBSOT4* gene was preliminarily verified by VIGS technology. The metabolomics showed that the resistance of *Gossypium barbadense* to blight was related to the changes of flavonoid metabolites. This study provides new insights for the functional genomics of cotton and laid a foundation for further research on the molecular mechanism of *G. barbadense* resistance to FOV.

## 2. Results

### 2.1. Identification of Members of the SOT Gene Family

Local BLASTP program and HMMER 3.0 software were used to compare and retrieve the target files, and candidate sequences were obtained. The conserved domains were validated by online tools Pfam, SMART and CDD, and the protein sequences of *Arabidopsis* were used to identify four cotton species. In the end, 241 *SOT* genes were identified (Appendix A), including 46 genes in *G. arboreum* (A2), 44 genes in *G. raimondii* (D5), 77 genes in *G. hirsutum* (AD1), and 74 genes in *G. barbadense* (AD2). Notably, the number of *G. raimondii* (D5) in subgroup D was two fewer than that of *G. arboreum* (A2) in subgroup A. Similarly, the number of *SOT* genes in *G. hirsutum* was three more than that in *G. barbadense*. The number of *SOT* family members in the tetraploid cotton (*G. hirsutum*, *G. barbadense*) is less than the sum of the diploid cotton (*G. arboreum*, *G. raimondii*), indicating that there was likely a phenomenon of gene loss during the evolution of the species. These genes were renamed *GaSOT1-GaSOT46*, *GBSOT1-GBSOT74*, *GHSOT1-GHSOT77*, and *GrSOT1-GrSOT44*, based on their location on the chromosome (Appendix A).

Then, the amino acid sequences of *SOT* genes family members in four cotton species were analyzed. The amino acid length of *SOT* family members was 60–672 amino acid residues, the average sequence length was 281 amino acids, the molecular weight was 7.21–76.15 kDa, and the average amount was 32.41 kDa. pI ranges from 4.52 to 9.82, with an average of 6.53 (Appendix A).

### 2.2. Construction and Analysis of the Evolutionary Tree of SOT Family Genes

MEGA software v7.0 was applied to construct the phylogenetic tree of 241 SOT protein sequences in four cotton species, with a view to analyzing the evolutionary relationships among members of the *SOT* genes family (Figure 1). Finally, 241 members of the *SOT* family were divided into five different subfamilies, namely Cluster1–Cluster5. Cluster5 has the largest number of 61 genes, followed by Cluster1 and Cluster4 with 58 genes and 52 genes, respectively, and Cluster3 with 34 genes. In Cluster1, Cluster4 and Cluster5, the number of *G. barbadense* is one more than the number of *G. hirsutum*, and in Cluster2, the number of *G. barbadense* is six less than the number of *G. hirsutum*. The amount of *G. hirsutum* and *G. barbadense* in Cluster3 is the same. Interestingly, Cluster1-Cluster3 has more *G. arboreum* than *G. raimondii*, while Cluster4-Cluster5 has the opposite. It is worth noting that diploid *G. arboreum* and *G. raimondii* are always clustered with tetraploid *G. hirsutum* and *G. barbadense*, which also confirms that tetraploid *G. hirsutum* and *G. barbadense* are evolved by crossing diploid *G. arboreum* and *G. raimondii* [24].

### 2.3. Chromosomal Distribution of SOT Family Genes in Four Cotton Species

In order to better understand the chromosomal distribution of *SOT* family genes on four cotton species, we conducted visual analysis of these genes on chromosomes, and 241 *SOT* family genes are present on most chromosomes of four cotton species (Figure 2). In *G. hirsutum*, 77 genes were distributed on 22 chromosomes, and the number of *SOT* genes on each chromosome ranged from one to nine, among which D03 chromosome had the largest number of genes with nine. In *G. hirsutum*, there are 38 genes in subgenome A and 36 genes in subgenome D, and the number of genes in subgroup A is two more than that in subgroup D. However, there was no distribution of *SOT* genes in A03, A08, D02, and D08. The distribution of *SOT* genes in *G. barbadense* was slightly different from that in *G. hirsutum*. A total of 74 genes were randomly distributed on 21 chromosomes, and there were one to nine *SOT* genes on each chromosome. There was no distribution of *SOT* genes in A03, A06, A08, D02 and D08. In *G. arboreum*, 46 *SOT* genes were distributed on 11 chromosomes except Chr03 and Chr08, and the number of genes on Chr05 was the highest with 11, which was followed by Chr11 with 10 genes. Similarly, in *G. raimondii*, 44 *SOT* genes are distributed on 11 chromosomes except Chr03 and Chr08. Unlike *G. arboreum*, Chr05 has the highest number of *SOT* genes with nine, which was followed by Chr11 with eight.

### 2.4. Analysis of Gene Structure, Protein Motifs and Cis-Acting Elements

In order to further understand the possible structural evolutionary relationships of *SOT* family members of *G. barbadense*, we constructed phylogenetic trees for 74 *SOT* genes of *G. barbadense* using the NJ method, and we conducted motif association analysis and gene structure analysis (Figure 3). The phylogenetic tree and gene structure information were constructed by using the protein sequences and annotation files of 74 *SOT* members. Using MEME and TBtools to analyze conserved motifs in SOT proteins, we found a total of 20 motifs among 74 members of *G. barbadense*. The number of motifs in each group member is different, and each group has a similar motif composition. motif11 is present in most *SOT* genes, which is followed by motif10 and motif12. Among them, *GBSOT30*, *GBSOT64* and *GBSOT74* in group 3 contained the most motifs with 15 species. It is worth noting that *GBSOT1* in group 1 contained only motif5, indicating that these genes may have undergone mutations during evolution. To further explore the gene structure of the *SOT* family members of *G. barbadense*, we analyzed the intron–exon structure characteristics. As shown in Figure 3, the same group family has a similar arrangement of intron exons, among which *GBSOT28* contains the most exons in group 1 with eight. Next, *GBSOT38* and *GBSOT12* in group 1 have six exons with *GBSOT25* and *GBSOT60* in group 3.

In order to better investigate the mechanism of *SOT* genes regulation, we used the PlantCARE database to predict the *cis*-acting elements of the 2000 bp promoter region upstream of 74 *SOT* genes in *G. barbadense*. In *G. barbadense* (Figure 3), there are *cis*-acting elements involved in drought induction with MYB binding sites, and plant hormone-related *cis*-acting elements include abscisic acid response elements, salicylic acid response elements, MeJA response elements and auxin-responsive elements. The regulatory element was involved in seed-specific regulation, the *cis*-acting element was involved in low-temperature responsiveness, the regulatory element was involved in zein metabolism regulation, and it was also the element involved in defense and stress responsiveness. Through the analysis of the promoter, this will help us to verify the subsequent gene function.

### 2.5. Gene Replication and Collinearity Analysis

To study the evolutionary process of the species, we identified the repetitive types of *SOT* genes in four cotton species (Appendix A). Among the 46 *SOT* genes of the two diploid cotton species, 11 genes were Dispersed, 12 genes were Proximal, 9 genes were Tandem, and 4 genes were WGD (Whole Genome Duplication) or Segmental. Among the 44 *SOT* genes of *G. raimondii*, 12 genes were Dispersed, 9 genes were Proximal, 21 genes were Tandem, and 2 genes were WGD or Segmental. However, in tetraploid cotton, most of the 77 *SOT* genes of *G. hirsutum* were WGD or Segmental, including 34 genes, followed by Tandem, including 22 genes, while 13 genes were Dispersed genes, and only 8 genes were Proximal genes. Similar to *G. hirsutum*, most of the 74 *SOT* genes in *G. barbadense* were WGD or Segmental, including 36 genes, followed by Tandem, including 18 genes, while 12 genes were Dispersed, and only 8 genes were Proximal.

We analyzed collinearity analysis between tetraploid *G. hirsutum* subgroup A and subgroup D (Figure 4c), and we found 34 orthologous/paralogous pairs, which were grouped among *G. barbadense* (Figure 4d). A total of 33 orthologous/paralogous pairs were identified. In addition, a collinearity analysis in *G. arboreum* was analyzed (Figure 4a), and three orthologous/paralogous pairs were identified. Four orthologous/paralogous pairs were found in *G. raimondii* (Figure 4b).

In addition, multicollinearity analysis of the *SOT* gene was performed in *G. hirsutum* (AD1), *G. barbadense* (AD2), *G. arboreum* (A2), and *G. raimondii* (D5) (Figure 4e). We found that *G. hirsutum* and *G. arboreum* had 57 orthologous gene pairs, *G. barbadense* and *G. arboreum* had 54 orthologous gene pairs, *G. barbadense* and *G. hirsutum* had 56 orthologous gene pairs, *G. barbadense* and *G. raimondii* had 51 orthologous gene pairs, and *G. hirsutum* and *G. raimondii* had 48 orthologous gene pairs. Therefore, we speculate that the main reasons for the amplification of *SOT* family genes during evolution are whole genome replication events and fragment replication events in four cotton species.

### 2.6. Gene Replication and Collinearity Analysis

To study the differentiation mechanism of *SOT* genes in cotton polyploid repeat events, we calculated the non-synonymous and synonymous substitution ratios (Ka/Ks ratio) to discover the type of selection pressure of these homologous gene pairs during evolution (Appendix A). We calculated the Ka/Ks ratios of 340 homologous gene pairs in four cotton species. The Ka/Ks ratios between *G. arboreum* and *G. arboreum* were all less than 0.5, and the Ka/Ks ratios between *G. raimondii* and *G. raimondii* were all less than 0.5. However, among tetraploid *G. hirsutum*, there were 24 homologous gene pairs whose Ka/Ks ratio was less than 0.5 and eight pairs whose Ka/Ks value was between 0.5–0.99. Among *G. barbadense*, there were 23 homologous gene pairs whose Ka/Ks ratio was less than 0.5 and nine pairs whose Ka/Ks value was between 0.5 and 0.99. These results suggest that these *SOT* genes have undergone strong purification selection during evolution. However, the Ka/Ks ratio of one homologous gene pair in *G. barbadense* was greater than one, indicating the positive selection effect of this homologous gene pair, indicating that these genes have evolved rapidly in recent years and may have important significance for the evolution of species (Figure 4f).

In addition, the Ka/Ks ratios between 57 pairs of *G. arboreum* and *G. hirsutum* and 54 pairs of *G. arboreum* and *G. barbadense* were calculated. Among them, 34 homologous gene pairs between *G. arboreum* and *G. hirsutum* had a Ka/Ks ratio of less than 0.5, 18 homologous gene pairs had a Ka/Ks value between 0.5 and 0.99, and 3 homologous gene pairs had a Ka/Ks ratio that was greater than 1. Also between *G. arboreum* and *G. barbadense*, 34 homologous gene pairs had a Ka/Ks ratio that was less than 0.5, 16 homologous gene pairs had a Ka/Ks value between 0.5 and 0.99, and 1 homologous gene pair had a Ka/Ks ratio that was greater than 1. The Ka/Ks ratio of 35 homologous gene pairs between *G. hirsutum* and *G. barbadense* was less than 0.5, and the Ka/Ks value of 19 homologous gene pairs was between 0.5 and 0.99. The Ka/Ks ratio of 26 homologous gene pairs was less than 0.5, the Ka/Ks value of 22 homologous gene pairs was between 0.5 and 0.99, and the Ka/Ks ratio of 2 homologous gene pairs was greater than 1. Similarly, the Ka/Ks ratio of 30 homologous gene pairs between *G. hirsutum* and *G. raimondii* was less than 0.5, the Ka/Ks ratio of 17 homologous gene pairs was between 0.5 and 0.99, and the Ka/Ks ratio of 1 homologous gene pair was greater than 1, indicating that most *SOT* genes had undergone strong purification selection in the evolutionary process.

### 2.7. Tissue Expression Patterns of SOT Genes in G. barbadense

After examining closely related gene expression patterns and their function, in order to discuss the amount of members of the family of *SOT* gene expression in different tissue changes, we first used the organization of the transcriptome data [25] and analyzed their expression quantity changes (Figure 5a), finding *GBSOT5*, *GBSOT36*, and *GBSOT51*. The expression levels of *GBSOT67* in the root and pistil were slightly higher than those in other tissues. It is worth noting that the expression of *GBSOT17* in the stem and leaf is the highest among all *SOT* genes (Figure 5b). Similarly, the expression of *GBSOT33* in the petal, calycle, stamen and pistil was the highest among all *SOT* genes (Figure 5c).

In addition to exploring the expression of *SOT* genes in tissues, we also studied the expression changes of the ovule and fiber transcriptome data. The expression levels of *GBSOT17* and *GBSOT51* at 25DPA (days post-anthesis) of fiber development were higher than those of other genes, suggesting that these genes may be involved in the middle and late stages of *G. barbadense* fiber development. Similarly, the expression levels of *GBSOT17* and *GBSOT72* in 5DPA ovule development were higher than those of other genes, indicating that these genes may be involved in the early stage of *G. barbadense* fiber development (Figure 5d).

In addition, we also used the root transcriptome data of Xinhai14 [21], a susceptible variety, and 06-146, a resistant variety, 40 h after inoculation to explore the changes in the expression of *SOT* family members. The results showed that the expression levels of most *SOT* genes did not change significantly before and after infection, but the expression levels of *GBSOT17* and *GBSOT33* were significantly different in disease-resistant and disease-susceptible materials and super-susceptible and super-resistant materials, respectively. These two genes play an important role in the FOV resistance of *G. barbadense* (Figure 5e).

### 2.8. Expression Patterns of SOT Genes in G. hirsutum

In addition to exploring the changes of *SOT* genes expression in *G. barbadense*, we also investigated the expression of *SOT* genes in the tissue [25], ovule and fiber development of *G. hirsutum*. In *G. hirsutum* tissues (Figure 6a), the expression of *GHSOT14* in the root was higher than that of other genes, which was followed by *GHSOT77*. The expression of *GHSOT14* in calycle was higher than that of other genes, which was followed by *GHSOT37*. Meanwhile, the expression of *GHSOT14* is higher than that of other genes in the stamen, pistil and receptacle. In addition, during the development of cotyledon, the expression of *GHSOT14* was the highest at 24 h, which was followed by 48 h (Figure 6b). In the development process of seeds, the expression level gradually increased with the increase in time, and the expression level reached the highest at 10 h. Interestingly, the expression level of *GHSOT37* was the highest at 0 h, and it gradually decreased with the increase in time, reaching the lowest level at 10 h. During root development, the expression of *GHSOT54* is higher than that of other genes. In the process of fiber development (Figure 6c), *GHSOT14* had a higher expression level at 20 DPA during ovule development than other genes, indicating that this gene may be involved in fiber development in the middle and late period of *G. hirsutum*. During fiber development, the expression of *GHSOT75* was the highest at 3 DPA during ovule development, followed by 5 DPA and 10 DPA during fiber development, suggesting that this gene may be involved in the early fiber development of *G. hirsutum*. Interestingly, by using the transcriptome data of Sicala V-2 and CSS386 (Figure 6d), we found that *GHSOT75* had a higher expression level than other genes at 4 DPA and 6 DPA during the development of fuzz fiber, indicating that this gene may be involved in the development of fuzz fiber [26].

To investigate the mechanism of the *SOT* gene response to abiotic stress, we used RNA-seq data to analyze changes in gene expression levels under cold, heat, salt, and PEG stress (Figure 6e) [25]. The results showed that under cold stress, the expression level of *GHSOT8* began to increase at 1 h, reached the peak at 3 h, and then decreased at 6 h. At the same time, under the four abiotic stress conditions, the expression level of *GHSOT14* was higher than that of other family members. Compared with the control group, under heat stress, the expression level of *GHSOT14* began to increase at 1 h, and it continued to rise with the increase in time, reaching the peak at 12 h. In addition, we also explored the expression level of the *SOT* gene in *G. hirsutum* based on the transcriptome data of cotton roots under the stress of *Verticillium dahliae* (Figure 6f) [25]. The results showed that the expression level of *GHSOT71* increased with the increase in time, and the expression level of *GHSOT71* at 48 h and 72 h was higher than that at other periods, indicating that this gene may be involved in the response of cotton to the treatment of *Verticillium dahliae*. The expression of *GHSOT54* reached the highest level at 6 h after infection by *Verticillium dahliae*, and then, it began to decrease gradually, suggesting that this gene may be involved in the early response of cotton to *Verticillium dahliae* treatment.

Cottonseed cannot be fully utilized as a food source due to the presence of toxic gossypol in the pigment glands of cottonseed. Therefore, it is of great significance to study the formation of pigment glands for the utilization of cotton seeds. We used the true leaf transcriptome data of *G. hirsutum* varieties L7 and Z17 with glanded material and *G. hirsutum* varieties L7XW and Z17YW with glandless material (Figure 6g) [27], and we found that the expression level of *GHSOT14* was relatively high in all the four materials. In addition, the expression level of *GHSOT77* was relatively high in L7 and L7XW. These two factors may regulate the development of pigment glands in *G. hirsutum*. In addition, based on the transcriptome data of TDZ (Thidiazuron) sensitive materials CIR12 and CCIR50 (Figure 6h) [28], we also found that the expression levels of *GHSOT37* and *GHSOT71* were 3.8 times and 12 times that of the control group at 48 h after TDZ treatment, respectively, indicating that these two genes may be involved in the response of cotton to TDZ treatment. Interestingly, we also found that the expression level of *GHSOT77* at 15 °C (low temperature) and 144 h after TDZ treatment was six times higher than that after TDZ treatment; at 25 °C (normal temperature), the expression level of *GHSOT77* was 21 times higher than that after TDZ treatment. These results indicated that this gene may be involved in the response of cotton to TDZ treatment at room temperature and low temperature.

### 2.9. Analysis of SOT Gene in G. barbadense by qRT-PCR

Our previous analysis showed that the expression of a subset of *SOT* genes changes significantly under FOV stress. Based on transcriptome data analysis and previous QTL mapping results [21,23], we speculated that *GBSOT4*, *GBSOT17* and *GBSOT33* may play an important role in the resistance of *G. barbadense* to FOV. We used qRT-PCR to further study the expression patterns of these three genes under FOV stress in disease-resistant cultivar (06-146) and a susceptible cultivar (Xinhai14), respectively. The results showed (Figure 7) that except at 1 h, the expression level of *GBSOT4* in Xinhai14 was higher than that of 06-146, and the expression level of 06-146 in other periods was higher than that of Xinhai14 with significant differences, indicating that this gene may play a positive regulatory role in coping with FOV in *G. barbadense*. However, *GBSOT17* was on the contrary. Although the expression level of *GBSOT17* in 06-146 was higher than that of Xinhai14 at 12 h, the expression level of *GBSOT17* was lower than that of Xinhai14 at other periods, suggesting that *GBSOT17* may play a negative regulatory role in coping with the FOV of *G. barbadense*. The expression pattern of *GBSOT33* and *GBSOT4* is similar. The *SOT* genes described above showed significant changes at different time points after FOV stress, indicating that these genes may be involved in the process of FOV stress in *G. barbadense*.

### 2.10. Transcription Analysis SOT Members in G. barbadense

Although the sequence structure, collinearity analysis, and selection pressure analysis of *SOT* gene families have been thoroughly studied, their potential role in the FOV resistance of *G. barbadense* remains unclear. We collected published transcriptome data [21] containing 16 root samples (from *G. barbadense* materials Xinhai14 and 06-146 at age 40 h and before and after FOV with extreme trait materials). Xinhai14 belongs to the susceptible variety, while 06-146 belongs to the resistant variety. Finally, we screened 35,560 genes with FPKM > 1 for WGCNA (weighted gene co-expression network analysis).

By combining modules with similar expressions using the dynamic shear tree method for weight values, a total of 19 modules were obtained in Xinhai14 and 06-146 materials (Figure 8a). The turquoise module contained the most genes (9687 genes), while the light-yellow module contained the fewest genes. With only 111 genes, each module contains an average of 1871 genes.

For materials of Xinhai14 and 06-146, core modules were screened according to the standards of |r| > 0.60 and *p* < 0.01 (Figure 8b). Among them, we found that the MEturquoise module was significantly negatively correlated with susceptible materials before FOV, and it was significantly positively correlated with susceptible materials after FOV. The MEblue module was significantly positively correlated with susceptible materials before inoculation and negatively correlated with susceptible materials after FOV. The MEgreen modules were significantly positively correlated with the disease-resistant materials before FOV. Of the 74 *SOT* family members of *G. barbadense*, 14 *SOT* genes—*GBSOT4*, *GBSOT5*, *GBSOT9*, *GBSOT17*, *GBSOT19*, *GBSOT22*, *GBSOT30*, *GBSOT37*, *GBSOT39*, *GBSOT41*, *GBSOT50*, *GBSOT64*, *GBSOT68*, and *GBSOT74*—belong to the MEblue mode Block; while three *SOT* genes, *GBSOT44*, *GBSOT47* and *GBSOT55*, belong to the MEgreen module. In addition, 12 *SOT* genes—*GBSOT6*, *GBSOT12*, *GBSOT25*, *GBSOT28*, *GBSOT33*, *GBSOT38*, *GBSOT51*, *GBSOT56*, *GBSOT60*, *GBSOT67*, *GBSOT71* and *GBSOT72*—belong to the MEturquoise module. Among them, the MEblue module is mainly enriched in amino sugar and nucleotide sugar metabolism, MAPK signaling pathway plant, cysteine and methionine metabolism, plant hormone signal transduction and other pathways (Figure 8c). Meanwhile, the MEgreen modules are mainly enriched in homologous recombination, flavone and flavonol biosynthesis, phenylpropanoid biosynthesis and glutathione metabolism and other pathways (Figure 8d). In contrast, the MEturquoise module is mainly concentrated in the ribosome, glutathione metabolism, biosynthesis of amino acids, carbon metabolism, sesquiterpenoid and sesquiterpenoid triterpenoid biosynthesis and other pathways (Figure 8e). In conclusion, the above core modules may regulate the resistance mechanism of *G. barbadense* to FOV through plant hormones, signal transduction by multiple pathways, glutathione metabolism, the biosynthesis of flavonoids and flavonols, carbon metabolism, sesquiterpenoid and triterpenoid biosynthesis, and other pathways.

### 2.11. Functional Validation and Interaction Network Construction for GBSOT4

Based on the previous QTL mapping results, major disease-resistance genes were screened, and *GB_A01G0479* (*GBSOT4*) was identified as a candidate gene for resistance to FOV of *G. barbadense*. Subsequently, we conducted VIGS experiments on this gene; the silenced variety was 06-146, and the results showed the following. After 15 days of injection, the chlorophyll synthesis of pTRV2-CLA plants in the positive control group was blocked, and the albino phenotype appeared in both true leaves and the stem vein (Figure 9a). The silencing efficiency of the target gene was detected by qRT-PCR, and the expression of the target gene in plants in the experimental group was significantly decreased (Figure 9b). The target gene was silenced in the plant. After two true leaves were grown, cotton plants of the treated and control groups were impregnated by root soaking and inoculated in spore suspension with a spore concentration of 1 × 10^7^/mL^−2^ × 10^7^/mL for 30 min. The incidence and disease index were observed 15 days after inoculation. Compared with the control group, plants with silenced target genes showed more severe leaf yellowing, curling, blighting, thinning of stems, dwarfing and slow growth. The disease index of the control was 17.9, and that of the plants after silencing was 38.5, which was significantly higher than that of the control group (Figure 9c).

Although the functionality of *GB_A01G0479* has been preliminarily validated by VIGS technology, its network of interactions is unknown. We carried out the WGCNA analysis and found that *GB_A01G0479* belonged to the MEblue module, from which 267 genes with a weight value greater than 0.04 were selected as interacting genes with *GB_A01G0479* (Appendix A). In order to detect the potential role of the *GB_A01G0479* interaction network (Figure 9d), we performed KEGG analysis on these 267 genes, and we found that these genes were mainly enriched in some metabolite synthesis pathways (Figure 9e). This suggests that the accumulation of some metabolites may be involved in the resistance to wilt infection.

### 2.12. Overview of the Metabolomic Profiling

Principal component analysis (PCA) was performed on six samples, of which three were samples after normal plant inoculation, and the other three were samples after silent *GBSOT4* inoculation. The validity of sequencing results could be seen from the dispersion degree of samples. As can be seen from the results (Figure 10a), the three replicated data between CK control (normal plant infection) and VIGS-FOV (silencing *GBSOT4*-infected plants) were well grouped into one class, further demonstrating the accuracy of the experiment and also indicating a significant difference between the two. The analysis method of OPLS-DA (Orthogonal Partial Least Squares-Discriminant Analysis) can effectively show the correlation between the predicted principal component and the model. We also use Partial Least Squares-Discriminant Analysis (PLS-DA) to further estimate the metabolite difference between CK and VIGS-FOV. The results indicate that the current model can be used as a distinction between CK control and VIGS-FOV (Figure 10b).

We also counted the number of differential metabolites between the two groups, and a total of 113 differential metabolites were detected, of which 81 were up-regulated and 32 were down-regulated (Figure 10c).

### 2.13. Targeted Metabolomic Analysis of Flavonoids Metabolite Content

Further analysis of the differential metabolites between the two groups showed that the most up-regulated flavonoid metabolites were Eupatorin-5-methylether (3′-hydroxy-5,6,7,4′-tetramethoxyflavone), whose up-regulated ratio was 4.38 times. Naringenin-7-O-(2”-O-APiosyl)glucoside was followed by an up-regulation factor of 3.52 times. The flavonoid metabolite with the largest down-regulation was Naringenin-4′-O-glucoside* with a down-regulation factor of 2.41 (Figure 10d). These results suggest that the resistance of *G. barbadense* to FOV may be due to the increased content of these flavonoid metabolites (Appendix A).

### 2.14. KEGG Functional Annotation and Enrichment Analysis of Flavonoid Differential Metabolites

KEGG (Kyoto Encyclopedia of Genes and Genomes) is a common tool for combining plant metabolite content and gene expression information. This study further analyzed the differential metabolites enriched in the KEGG metabolic pathway (Figure 10e). The total flavonoid differential metabolites were enriched in five pathways as follows: flavonoid biosynthesis, biosynthesis of secondary metabolites, flavone and flavonol biosynthesis, isoflavonoid biosynthesis, and metabolic pathways. Among them, the biosynthesis of secondary metabolites was the pathway with the most differentially enriched metabolites, and 66.67% of DEMs (differential expression metabolites) was classified in this pathway (Figure 10f).

## 3. Discussion

FOV is the main disease of *G. barbadense*. In recent years, due to continuous planting, FOV has seriously affected the production of *G. barbadense*. Improving the disease resistance of *G. barbadense* is a difficult and complicated task. The further development of genetic engineering has brought great convenience to scientific researchers. The existence of cotton genome sequences has enabled researchers to explore the evolution and functional analysis of different gene families, many of which have been previously examined in cotton: for example, *GhIFR* [29], *GhGGPS* [30], *GhTBL* [31], *GhDREB* [32], *GhSBT* [33], *GhLOG* [34], *GhGATL* [35], *Gb_ANR* [36], and *GbCHI* [37].

Based on the previous QTL mapping results, we screened a major disease-resistance gene *GB_A01G0479*, which belongs to the *SOT* family. Subsequently, we conducted a systematic study of this gene family in four cotton species and determined the evolutionary relationships through phylogenetic analysis, gene structure, protein motif, chromosome localization, gene duplication, collinearity analysis, and selection pressure analysis. The role of the GBSOT gene was observed through the analysis of *cis*-elements, examination of tissue-specific expression patterns, and investigation of its response to abiotic stress. The function of the *GBSOT* gene was observed under FOV stress. Then, the VIGS technique was used to study the response mechanism of gene *GB_A01G0479* to pathogen infection in *G. barbadense*.

### 3.1. Basic Analysis of SOT Family Members of G. barbadense

We identified 241 *SOT* genes in four cotton species based on the published reference genome information from four cotton species. Then, we analyzed the amino acid sequences of four cotton family members. The amino acid length of *SOT* family members is 60–672 amino acid residues, the average sequence length is 281 amino acids, the molecular weight is 7.21–76.15 kDa, and the average amount is 32.41 kDa. pI ranges from 4.52 to 9.82, with an average of 6.53, indicating that these proteins are basic proteins. The proteins encoded by members of the *SOT* family have different physical and chemical properties, as well as different functions and regulatory mechanisms, but they all have stable SOT domains. In order to analyze the evolutionary relationships among members of the *SOT* family, we constructed a phylogenetic tree. It is worth noting that diploid *G. arboreum* and *G. raimondii* always cluster with tetraploid *G. hirsutum* and *G. barbadense*, which also confirmed that tetraploid *G. hirsutum* and *G. barbadense* evolved from diploid *G. arboreum* and *G. raimondii* [38]. We also analyzed the gene structure and conserved motifs of members of the *SOT* family, showing that the same clades have a similar gene structure and conserved motifs. The distribution of *SOT* genes in diploid and tetraploid cotton is roughly the same.

At the same time, we conducted a multicollinearity analysis in four cotton species, and we speculated that the main causes of gene amplification in the evolution of *SOT* family genes were genome-wide replication events and fragment replication events. At the same time, we also calculated the Ka/Ks ratio of 340 homologous genes, and the results suggest that most *SOT* genes underwent strong purification selection in the evolutionary process, and a few homologous genes had positive selection effect [39]. In addition, we also predicted the *cis*-acting elements of *SOT* genes in *G. barbadense*, including some *cis*-acting elements involved in drought induction with MYB binding sites, and plant hormone-related *cis*-acting elements including abscisic acid response elements, salicylic acid response elements, MeJA response elements, auxin-responsive elements, etc. The results suggest that *SOT* genes not only participate in a variety of signaling pathways but also participate in plant growth and development and defense responses [40]. We know that the predicted *cis*-acting elements in the promoters using upstream activating promoter sequences are only speculative hypothesis, and none are known to be functional as predicted until tested experimentally. Furthermore, gene expression regulation can also be controlled in other ways such as via promoters/transcription, so the absence of predicted *cis*-acting elements in the promoter region does not mean that other RNA/proteins cannot be controlled through other means (RNA stability, processing, protein regulation, etc.).

### 3.2. Analysis of Expression Patterns of SOT Family Members in G. barbadense

The expression profile of the SOT gene may be strongly correlated with its functionality. We analyzed the expression pattern of *SOT* genes based on published transcriptome data. We found that the expression levels of some *SOT* genes were higher in the tissues of *G. barbadense*, among which *GBSOT5*, *GBSOT36*, *GBSOT51* and *GBSOT67* were slightly higher in the root and pistil than in other tissues, respectively, and *GBSOT17* was the highest in the stem and leaf among all *SOT* genes. Similarly, the expression of *GBSOT33* in the petal, calycle, stamen and pistil was the highest among all *SOT* genes. In addition to tissue, we found that *GBSOT17* and *GBSOT51* had higher expression levels at 25 DPA of fiber development than other genes, suggesting that these genes may be involved in fiber development in the middle and late stages of *G. barbadense*. Similarly, *GBSOT17* and *GBSOT72* may participate in the fiber development of *G. barbadense* in the early stage. In addition, we also found significant differences in the expression levels of *GBSOT17* and *GBSOT33* in disease-resistant materials and disease-susceptible materials as well as super-susceptible materials and super-resistant materials, respectively, suggesting that these two genes play an important role in the FOV resistance of *G. barbadense*.

In addition to the changes of *SOT* gene expression in *G. barbadense*, the expression of *SOT* genes was also investigated during the development of the tissue, ovule and fiber of *G. hirsutum*. *GHSOT14* and *GHSOT14* are specifically expressed in the root and calycle, respectively. During fiber development, the expression of *GHSOT14* at 20 DPA during ovule development was higher than that of other genes, suggesting that *GHSOT14* might be involved in fiber development in the middle and late period of *G. hirsutum*. At the same time, *GHSOT75* may be involved in the initial fiber development of *G. hirsutum*. Interestingly, we found that the expression of *GHSOT75* was higher at 4 DPA and 6 DPA in fuzz fiber development than other genes, indicating that this gene may be involved in the development of fuzz fiber in *G. hirsutum*. In addition to tissue, we also investigated the response mechanism of *SOT* genes to abiotic stress, and we found that *GHSOT8* and *GHSOT14* may be involved in the response of several abiotic stresses. In addition, we also found that *GHSOT71* and *GHSOT54* may be involved in the response of late and early cotton to the treatment of *Verticillium dahliae*, respectively. At the same time, we used the true leaf transcriptome data of glanded varieties L7 and Z17 and glandless varieties L7XW and Z17YW, and we found that *GHSOT14* and *GHSOT77* may regulate the development of pigment glands in *G. hirsutum*. *GHSOT37*, *GHSOT71* and *GHSOT77* may be involved in the response of cotton to TDZ treatment.

Then, we used qRT-PCR to detect the expression patterns of *GBSOT4*, *GBSOT17* and *GBSOT33* under FOV stress, and the findings demonstrated that these genes exhibited noteworthy alterations at various time intervals following FOV stress, suggesting their potential involvement in the FOV stress process in *G. barbadense*. Through transcriptome data, we found that among the 74 *SOT* family members of *G. barbadense*, 14 *SOT* genes belonged to the MEblue module, 3 *SOT* genes belonged to the MEgreen module, and 12 *SOT* genes belonged to the MEturquoise module. KEGG analysis was performed on these three modules. It was found that these core modules may regulate the resistance mechanism of *G. barbadense* to FOV through plant hormones, signal transduction by multiple pathways, glutathione metabolism, the biosynthesis of flavonoids and flavonols, carbon metabolism, and sesquiterpenoid and triterpenoid biosynthesis.

### 3.3. Functional Verification and Metabolomics Analysis of GBSOT4

Based on the previous QTL mapping results, major disease-resistance genes were screened, and *GB_A01G0479* (*GBSOT4*) was identified as a candidate gene for resistance to FOV of *G. barbadense*. Subsequently, VIGS experiments were conducted on this gene, and the silencing efficiency of the target gene was detected when the positive control TRV2-CLA leaves showed an albino phenotype, and the results showed that the target gene was silenced. The plants silenced with the target gene and the control group were inoculated with FOV to explore the resistance of *GBSOT4* to *G. barbadense*. After inoculation, the results showed that the plants silenced with the target gene had more serious wilting, drying and cracking than the control group, and the disease index of the plants silenced with the target gene was significantly higher than that of the control group, indicating that *GBSOT4* may be involved in protecting the production of *G. barbadense* from FOV. At the same time, we analyzed the genes interacting with *GB_A01G0479* and carried out interaction network analysis, and we found that these genes were mainly enriched in some metabolite synthesis pathways. We speculated that when infected by FOV, they might resist through the accumulation of some metabolites.

Physiological and biochemical barriers play an important role in cotton defense. In order to prevent the harm of pathogenic bacteria, cotton has accumulated a variety of antibacterial compounds, including terpenoids, gossyphenols, flavonoids and tannins. These substances can effectively inhibit the infection of verticillium wilt [41]. In cotton, catechins and quercetin have been shown to directly inhibit the spore formation, spore germination and mycelium growth of fungal pathogens. After cotton was infected by pathogens, the catechin yield increased significantly, and the catechin content in resistant cotton was higher than that in susceptible cotton [42]. In addition, naringin, dihydroxyquercetin and rutin inhibit the in vitro growth and accumulation of flavonoids in cotton, resulting in enhanced disease resistance [43].

The silencing of biosynthetic genes can cause potential metabolic remodeling and influence cotton resistance [44]. In this study, in addition to the differences in these metabolites mentioned above, there are some other metabolites, including Eupatorin-5-methylether(3′-hydroxy-5,6,7,4′-tetramethoxyflavone), Naringenin-7-O-(2″-o-apiosyl)glucoside, and Genkwanin-6-C-(2″-o-apiosyl)glucoside A series of flavonoid metabolites, such as Tricin-7-O-(2″-feruloyl)glucuronide, were accumulated in response to pathogen infection of cotton.

## 4. Materials and Methods

### 4.1. Identification of Cotton SOT Gene Family Members

First, we downloaded the *G. arboreum* (ICR), *G. raimondii* (JGI), *G. hirsutum* (ZJU), *G. barbadense* (ZJU), and the reference genome and genome annotation information GFF3 file from the CottonFGD database (https://cottonfgd.org/, accessed on 1 August 2023) [45]. Local BLASTP was used to search for the *SOT* genes in four cotton species using *Arabidopsis* SOT protein sequences. The HMM (Hidden Markov Model) map obtained from the Pfam (PF00685) database was used to verify the identified *SOT* genes [46]. The domain information of the SOT protein was further confirmed by NCBI Batch-CDD search. In order to explore the physical and chemical properties of the *SOT* genes, using an online tool (http://cn.expasy.org/tools, accessed on 1 August 2023), we calculated the number of amino acid residues, relative molecular weight, and theoretical isoelectric point of the SOT protein in cotton [47].

### 4.2. Chromosomal Locations and Gene Replication Analysis of SOT Genes

In order to study the chromosomal positions of *SOT* genes in four cotton species, the GFF3 file of cotton genome annotation data was downloaded from the CottonFGD database (https://cottonfgd.org/, accessed on 1 August 2023) [45], and the physical chromosomal positions of all *SOT* genes in the four cotton species were visualized using TBtools software (v1.098769) [48]. In addition, MCScanX software (V1.1) was used to analyze the genomic collinear blocks and gene replication [49].

### 4.3. Construction of a Phylogenetic Tree of SOT Family Proteins

To explore the evolutionary correlation among the SOT genes of four cotton species (*G. arboreum*, *G. hirsutum*, *G. raimondii*, *G. barbadense*), we used MEGA (MEGA7) and ClustalW to perform the multiple sequence alignment of 241 obtained *SOT* genes [50]. Based on the comparison results, the evolutionary tree was constructed using the ML (maximum likelihood) method, and the bootstrap value was set to 1000.

### 4.4. Gene Structure and Conserved Protein Motif Analysis of SOT Family Genes

Then, we used TBtools software to visually analyze the MEME file, the NWK file of phylogenetic tree analysis and the GFF3 genome annotation file of *G. barbadense* [48].

### 4.5. Expression Pattern and Cis-Element Analysis of SOT Family Genes

First, *G. barbadense* varieties Hai7124 RNA-seq data (PRJNA490626) from NCBI (https://www.ncbi.nlm.nih.gov/, accessed on 1 August 2023) were downloaded. Using these data, we examined the *GBSOT* gene expression in eight distinct tissues (root, stem, leaf, petal, receptacle, calycle, stamen, pistil), as well as during various stages of fiber and ovule development, exploring pheatmaps of expression using TBtools software [48]. Then, we downloaded the RNA-seq data of *G. hirsutum* TM-1 under cold, heat, salt and PEG stress and analyzed them using TBtools software [48]. 

In order to preliminarily explore the function of gene expression regulation, we extracted the upstream 2.0 kb sequence of the start codon from *SOT* family genes of *G. barbadense* as a promoter sequence for *cis*-element analysis. We used PlantCARE (http://bioinformatics.psb.ugent.be/webtools/PlantCARE/html, accessed on 1 August 2023) for the further analysis of *SOT* gene promoter regions of *cis*-elements, and we used the TBtools software for *cis*-element information visualization [48].

### 4.6. Collinearity and Selective Pressure Calculation Analysis of SOT Family Genes

In order to explore the evolutionary relationship and selection pressure of four cotton *SOT* family genes, and to find collinearity genes in the whole genome, we blasted all cotton protein sequences and compared them with MCScanX software [49]. Finally, TBtools software was used for visual analysis [48], and the non-synonymous replacement (Ka) and synonymous replacement (Ks) rates of duplicate genes were calculated using TBtools software [48].

### 4.7. Weighted Gene Co-Expression Network Analysis

We collected the published transcriptome data of 16 root samples (after FOV injection, Xinhai14 and 06-146 and extreme character materials from 40 h) for WGCNA analysis, and we utilized the R program’s WGCNA software (V4.1.1) package [51] to construct a weighted gene co-expression network. A total of 18 transcriptome samples were obtained, and the standardized gene expression matrix was used as input. Following threshold screening, we selected β = 5 as the power for the original scaled relation matrix, resulting in the acquisition of the unscaled adjacency matrix. The minimum number of genes within a module is set at 30 (minModuleSize = 30).

### 4.8. KEGG Enrichment Analysis and Interaction Network Construction

The genes within the target module were subjected to KEGG enrichment analysis using TBtools [48]. The thresholds for significance were set at *p* < 0.01 and Q < 0.05. Additionally, to gain insights into the potential interaction network of core genes, we computed the Pearson correlation coefficient as the interaction weight between the target gene and the candidate gene. Then, we visualized the input Cytoscape (v.3.7.1) of the interactive network construction results.

### 4.9. qRT-PCR Analysis

The results of preliminary QTL mapping and transcriptome data showed that three *SOT* genes, *GBSOT4*, *GBSOT17* and *GBSOT33*, which may be involved in disease resistance, were extracted from the stem RNA of 06-146 and Xinhai14 after FOV injection, and their expressions were detected at different periods. Three biological replications and three technical replications were performed, and the experiment was amplified on a real-time fluorescence quantifier 7500. The relative expression of genes was analyzed by the 2^−ΔΔt^ method [52]. Primers were used in this study (Appendix A).

### 4.10. Virus-Induced Gene Silencing of GBSOT4 and Metabolomic Analysis

The *GBSOT4* clone plasmid was used as the template to amplify the non-conserved region of the target fragment 447 bp. The target fragment was connected to the expression vector pTRV2 by the homologous recombination method. The connected products are converted into *E. coli* cultures and sent for sequencing in order to confirm the correct structure with the correct/precise sequence. OD600 was adjusted to 0.7 with aseptic suspensions containing 10 mmol/LMES, 20 μmol/L acetosyringone and 10 mmol/LMgCl_2_ (pTRV1:pTRV2 unloaded; pTRV1:pTRV2-CLA; pTRV1:pTRV2-*GBSOT4* was mixed according to 1:1) and left for more than 3 h away from light [20]. The leaves were slightly bruised by acupuncture and slowly injected with a syringe until the two cotyledon leaves were completely soaked in water. The injected cotton seedlings were cultured without light for 24 h; then, they were cultured in light at 25 °C for 16 h and dark at 23 °C for 8 h. After 15 days of Fov7 inoculation, the occurrence of malgrowth plants was monitored [33]. A minimum of 15 seedlings were subjected to treatment in each trial, and the disease severity index (DSI) was computed for each individual seedling [53].

For metabolomics measurements, samples of fresh cotton stems were freeze-dried and ground into powder. The metabolites were then extracted using 1 mL of 80% (*v*/*v*) methanol per 100 mg of dried sample at 4 °C overnight, which was sheltered from light. The extract supernatant was filtered by a 0.22 μm microporous filter SCAA-104 (ANPEL, Shanghai, China). The metabolites were analyzed by a LC-ESI-MS-MS system (Ultra Performance Liquid Chromatography, UPLC) (ExionLC^TM^ AD, https://sciex.com.cn/, accessed on 1 August 2023) and (Tandem mass spectrometry, MS/MS). Qualitative analysis of primary and secondary mass spectrometry data was performed based on the public database (MassBank) and the self-built MetWare database (accessed on 1 August 2023) [54,55]. Metabolites with multiple changes of >2 or <0.5 were considered significant in this study.

## 5. Conclusions

In this study, the *SOT* gene families of four cotton species were comprehensively analyzed, and for the first time, an analysis was conducted on the structural characteristics, phylogenetic relationships, gene structures, expression patterns, evolutionary relationships, selection pressure analysis, and stress response of the *SOT* members in *G. barbadense*. A total of 74 *SOT* genes were identified. According to the phylogenetic tree, the SOT protein sequences in four cotton species were divided into five different subfamilies. The physical locations of these genes on chromosomes were also mapped, the *cis*-acting elements of the *SOT* gene in *G. barbadense* were predicted, and some *cis*-acting elements related to plant hormones were found. We identified and observed the repeated types of *SOT* genes in four cotton species. In tetraploid gossypium, most belong to WGD (Whole Genome Replication) or fragment replication; in diploid gossypium, only a few belong to WGD or Segmental and Dispersed. We calculated the Ka/Ks ratio of the *SOT* homologous genes and explained the selective pressure between *SOT* genes. Through the analysis of *SOT* gene expression patterns, we found some *SOT* genes that were specifically expressed in the specific tissue or fiber development period in *G. barbadense*. 

We used qRT-PCR to detect the expression of *GBSOT4*, *GBSOT17* and *GBSOT33* under FOV stress, and we used WGCNA analysis to explore the mechanism of *SOT* gene regulation against FOV in *G. barbadense*. In addition, we conducted a VIGS experiment on *GBSOT4*; furthermore, the findings demonstrated a substantial increase in the disease index of plants undergoing gene silencing compared to the control group. This indicates that *GBSOT4* might play a crucial role in defending against the occurrence of FOV in *G. barbadense*. Subsequent metabolomics analysis showed that the accumulation of some flavonoid metabolites, such as Eupatorin-5-methylether (3′-hydroxy-5,6,7,4′-tetramethoxyflavone, was in response to pathogen infection of cotton to resist the occurrence of FOV. This study provides useful information for the evolution of the cotton *SOT* gene family and the biological function of *GBSOT*, and it lays a foundation for further study of the function of *GBSOT* in cotton.

## Figures and Tables

**Figure 1 plants-12-03529-f001:**
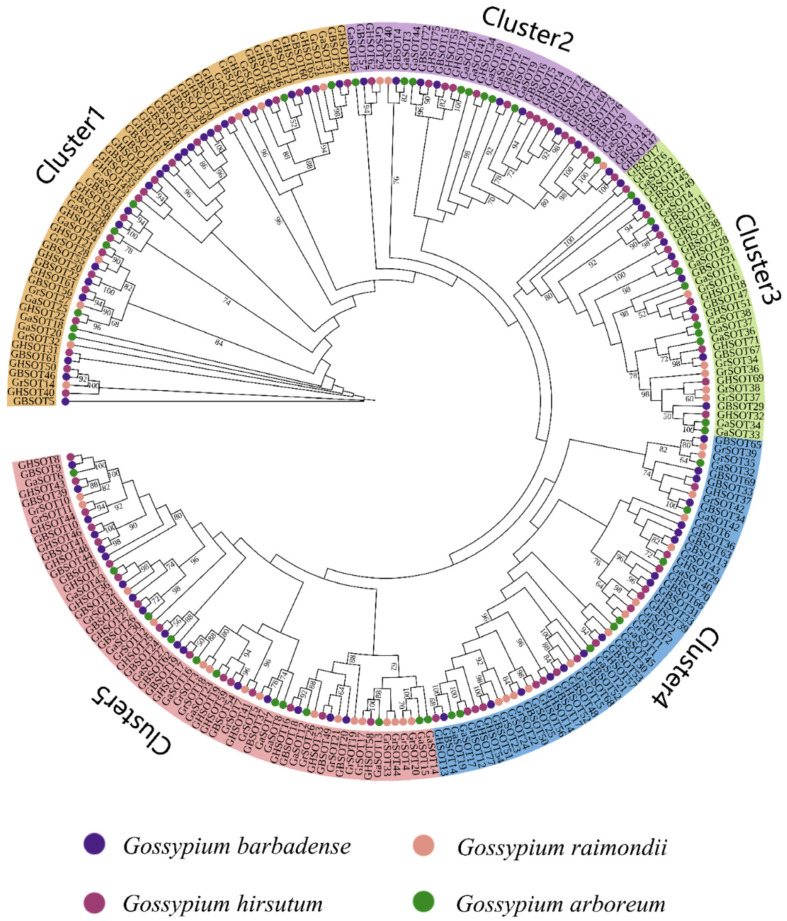
Phylogenetic analysis of *SOT* members in four cotton species.

**Figure 2 plants-12-03529-f002:**
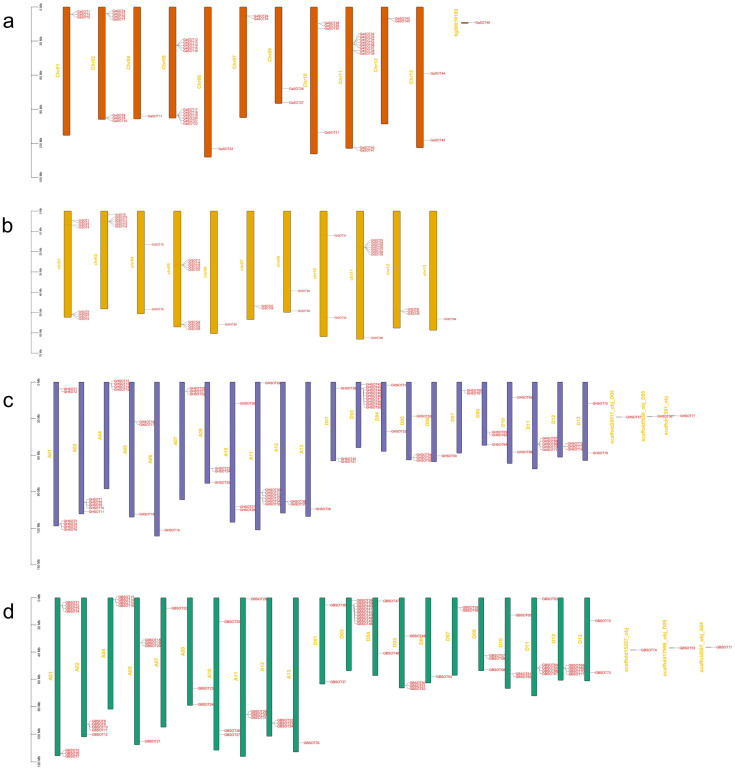
The distribution of the *SOT* gene on chromosomes of four cotton species.(**a**) *G. arboreum*. (**b**) *G. raimondii*. (**c**) *G. hirsutum*. (**d**) *G. barbadense*.

**Figure 3 plants-12-03529-f003:**
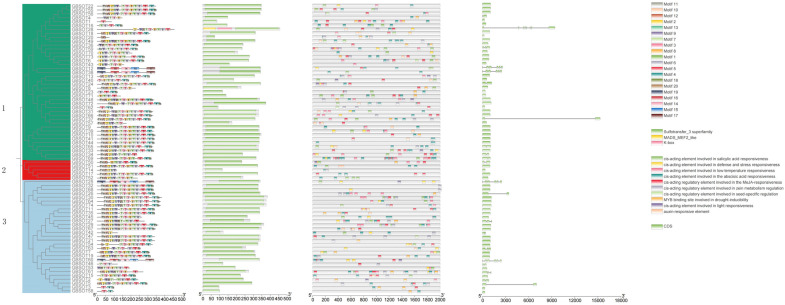
Phylogenetic tree of *GBSOT* gene, divided into three groups. motif composition and distribution, conserved domain, *cis*-acting elements of promoter, gene structure (exon–intron organization) from left to right.

**Figure 4 plants-12-03529-f004:**
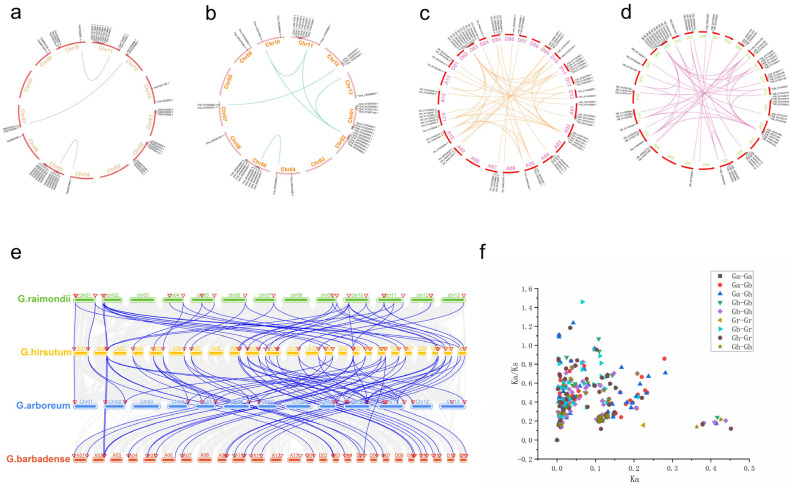
Collinearity analysis of *SOT* genes. (**a**) Collinearity analysis of *G. arboreum*. (**b**) Collinearity analysis of *Gossypium raimondii*. (**c**) Collinearity analysis of *Gossypium hirsutum*. (**d**) Collinearity analysis of *G. barbadense*. (**e**) Multiple synteny analysis among cotton *SOT* genes. Multiple synteny analysis was used to show the orthologous relationship among *G. arboreum*, *G. hirsutum*, *G. raimondii* and *G. barbadense* of *SOT* genes. The chromosomes of different kinds of cotton are shown in different colors. (**f**) Selection pressure analysis of the *SOT* gene family during evolution.

**Figure 5 plants-12-03529-f005:**
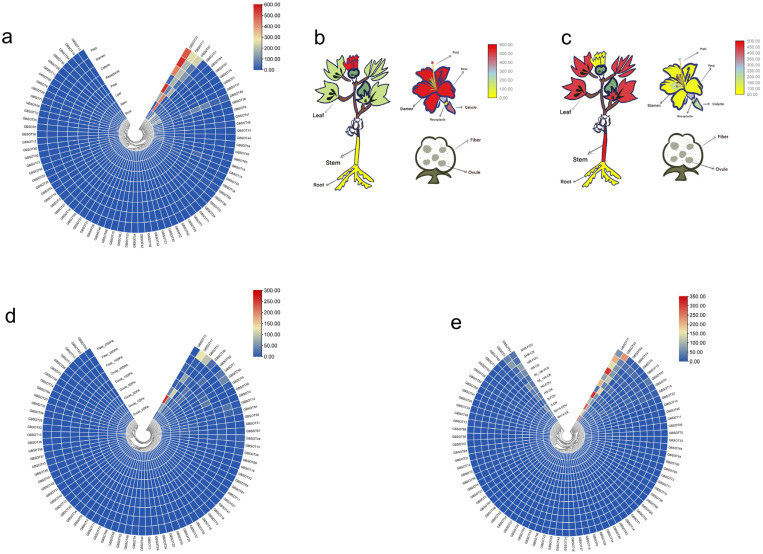
Expression patterns of *GBSOT* genes in different tissues. (**a**) The expression pattern of *GBSOT* genes in different organs (root, stem, leaf, petal, receptacle, calycle, stamen, pistil). (**b**) Pheatmap of *GBSOT17* gene in different tissues of cotton. (**c**) Pheatmap of *GBSOT33* gene in different tissues of cotton. (**d**) Expression patterns of *GBSOT* in ovule and fiber (0, 1, 3, 5, 10, 20 and 25 DPA). (**e**) Expression patterns of the *GBSOT* genes from *G. barbadense* under of FOV stress (0, 6, 12, 24, 48, and 72 h).

**Figure 6 plants-12-03529-f006:**
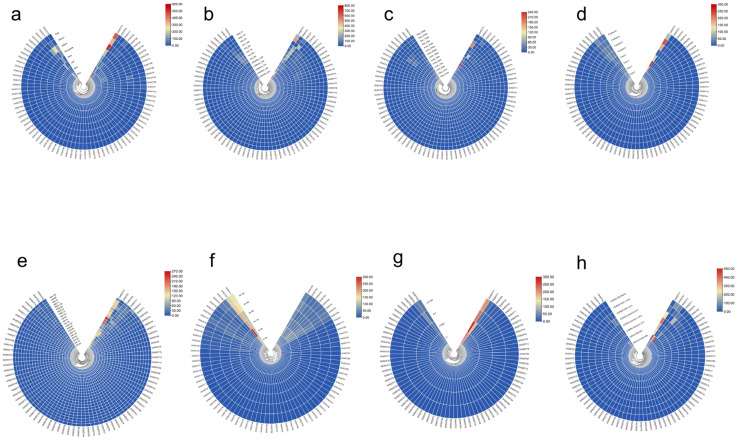
Expression patterns of *GHSOT* genes in different tissues. (**a**) The expression pattern of *GHSOTs* in different organs (root, stem, leaf, petal, receptacle, calycle, stamen, pistil). (**b**) Expression pattern of seed, cotyledon and root growth and development in upland cotton at different stages. (**c**) Expression patterns of *GHSOTs* in ovule and fiber (−3, −1, 0, 1, 3, 5, 10, 20, 25, and 35 DPA). (**d**) Expression levels of *GHSOTs* in ovule and fiber in fuzz material and fuzzless material at different periods (0, 2, 4, 6) DPA. (**e**) Expression patterns of *GHSOT* genes under cold, hot, salt and drought stress (0, 1, 3, 6, and 12 h). (**f**) Expression patterns of the *GHSOT* genes from G. hirsutum under *V. dahliae* stress (0, 6, 12, 24, 48, and 72 h). (**g**) Expression pattern of *GHSOTs* in glanded and glandless materials of upland cotton. (**h**) Expression patterns of the *GHSOT* genes from *G. hirsutum* under TDZ treatment.

**Figure 7 plants-12-03529-f007:**
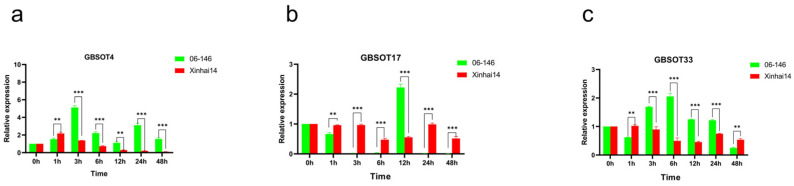
Analysis of the expression patterns of SOT family genes under FOV stress. (**a**) The expression of *GBSOT4* in the two materials after inoculation. (**b**) The expression of *GBSOT17* in the two materials after inoculation. (**c**) The expression of *GBSOT33* in the two materials after inoculation. The error bar represents the average ± SEs of three replicates. Compared with the control group, the difference was statistically significant, ** *p* < 0.01; *** *p* < 0.001.

**Figure 8 plants-12-03529-f008:**
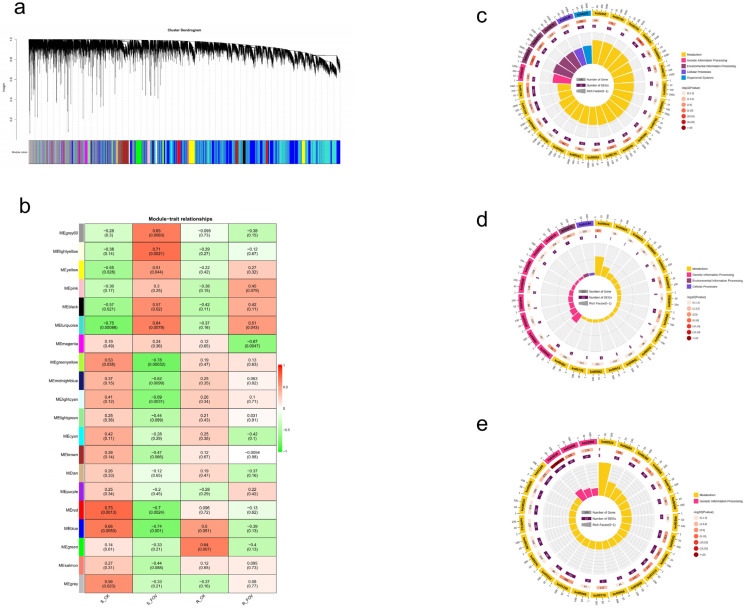
WGCNA for the transcriptome data in *G. barbadense*. (**a**) Gene cluster analysis results obtained using WGCNA. (**b**) Heatmap illustrating the correlation between modules and traits. The numbers within the squares denote correlation coefficients and *p*-values between modules. (**c**) KEGG pathway enrichment analysis conducted for the blue module. (**d**) KEGG pathway enrichment analysis performed for the green module. (**e**) KEGG pathway enrichment analysis carried out for the turquoise module.

**Figure 9 plants-12-03529-f009:**
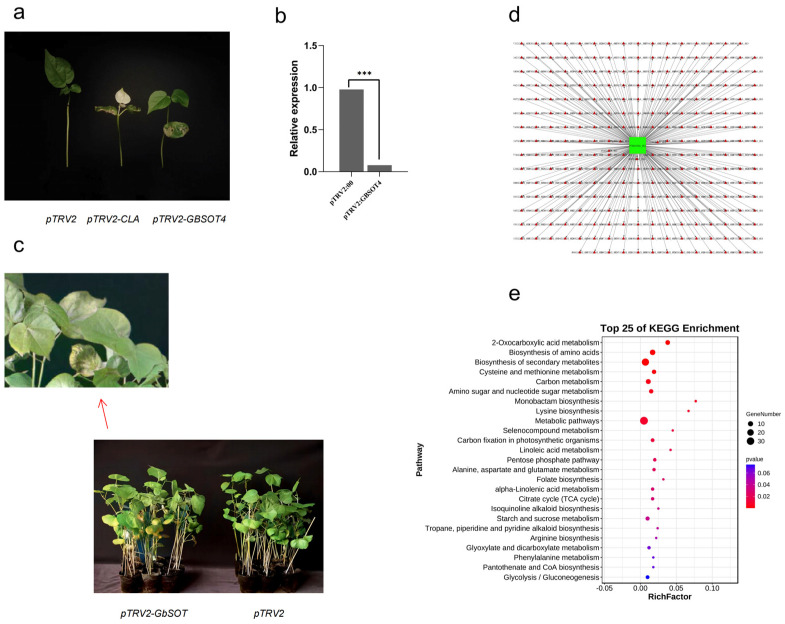
Functional verification of *GBSOT4*. (**a**) Phenotypic comparison of silent *GBSOT4* plants. (**b**) VIGS efficiency test of *GBSOT4* in *G. barbadense*. (**c**) Seedlings of control and silenced plants after inoculation with FOV at 15 days post-inoculation (dpi). (**d**) The entire network of *GBSOT4* constructed based on genome-wide transcriptome data. (**e**) The KEGG pathway enrichment analysis of the 267 genes. The error bar represents the average ± SEs of three replicates. Compared with the control group, the difference was statistically significant, *** *p* < 0.001.

**Figure 10 plants-12-03529-f010:**
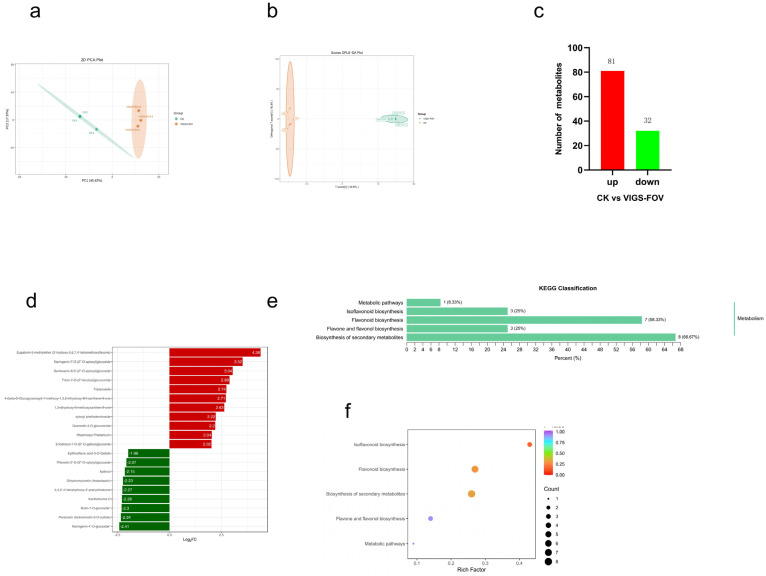
Metabolomic analysis of *GBSOT4* gene after inoculation with FOV. (**a**) Metabolite quantification was performed by principal component analysis (PCA) in two groups of samples. (**b**) The OPLS-DA analysis in two groups of samples. (**c**) Shows number of DEMs (up and down). (**d**) The fold change in the top 20 increased (red columns) or decreased (green columns) flavonoids. The fold change values were log2 transformed. (**e**) The KEGG classification of the identified DEMs. (**f**) Enrichment of KEGG pathway. The horizontal coordinate represents the enrichment factors of each biosynthetic pathway. The color of the point represents the *p*-value (color gradient from blue to red, representing low and high significance, respectively).

## Data Availability

The data presented in this study are available upon request from the corresponding author.

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
