# Peer review of "GBSOT4 Enhances the Resistance of Gossypium barbadense to Fusarium oxysporum f. sp. vasinfectum (FOV) by Regulating the Content of Flavonoid"

_plants, 2023, doi:10.3390/plants12203529_

Round 1

Reviewer 1 Report

The authors studied fourspecies of cotton, but it is not clear whether there is intraspecific variability? What genotypes were used? (lines, varieties, hybrids)

Author Response

Response to Reviewer 1 Comments

Dear Professor: Thank you very much for your comments on the manuscript. We have taken your suggestions in consideration point by point and made the updating of this revision.

Point 1: The authors studied fourspecies of cotton, but it is not clear whether there is intraspecific variability? What genotypes were used? (lines, varieties, hybrids)

Response 1: Thank you so much for your suggestion.We analyzed the SOT gene family using the reference genomes of four cotton species, and there were no such problems as the so-called hybrids

Thanks a lot for your patience and consideration.

Sincerely yours,

Jiao Yang

Postal address: College of Agriculture, Xinjiang Agricultural University, 311 Nongda East Road, Urumqi, 830052, China.

Telephone number: 18599052306

Fax number: 0991-8762263

E-mail address:[email protected]

Reviewer 2 Report

Plants Manuscript #:  2595713

Authors: Z. Su et al., 2023

Title:  GBSOT4 Enhances the Resistance of Gossypium barbadense to Fusarium ozysporum F. sp. vasinfectum (FOV) by Regulating the Content of Flavanoid

The authors have presented molecular genetic, transcriptomic, and metabolomic data about the predicted role of sulfotransferases, specifically GBSOT4 having a role resistance to critical cotton fungal pathogen Fusarium oxysporum.  Overall the information and experimental design and process seems solid.  However, there are some significant writing quality/clarity issues.  And, most importantly, none of the text in the figures were readable due to small and very-low resolution.  Thus, it was impossible to fully review this manuscript for as a reviewer the provided data in the figures was not verifiable due to the text in the figures being impossible to read.  This meant the read could not know what genes / proteins / labels went with what data and information.  For this reason, there a significant redesign / presentation of all figures is needed before a complete review is possible.

            Furthermore, a re-write/re-edit of the text is needed to correct many awkward/unclear sentences and inconsistent writing issues.  There were also many terms that needed to be defined and more information about some of the methods is needed.  I mention these in details in the detailed list below.

            Finally, in the section 2.13 (Lines 450-457) that address metabolomic changes related to FOV infection and possible resistance, it is not clear what, if any, of these have to do directly with SOT genes/proteins.  Many of the metabolites mentioned are not related at all, at least none that are described here, to SOT function.  The authors need to make a stronger an d more clear connection to how any of the metabolites with changes in abundance, including flavonoids, would relate to SOT genes/proteins and FOV resistance.  Otherwise, it is correlation and not necessarily causation.  Similar issue for Lines 586-602 in the Discussion.

Detailed Issues that need to be addressed.  Note, this is not necessarily all the issues, and once the text of the figures is re-done to readable, then there might be more issues that need to be addressed:

Abstract:

Line10:  Since “SOT” in this use is referring to proteins, it should NOT be italicized.  Later, when it refers to “SOT” genes, then it should (and usually correctly is) italicized.  Please check this issue throughout the manuscript, for often “SOT” is incorrectly italicized when also referring to proteins.

Line 15:  “… for the first time…” as it is stated currently implies it is the first time to “use bioinformatic methods…”. Suggest rewrite to clarify.

Line 21:  Depending on the journal, often “cis” is italicized, since it is a Latin term.  This is an issue throughout the manuscript.

Line 22:  Should be “four”, as numbers less than 10 should be written out in word form, as was correctly done in Line 18.

Line 26:  Define (write out fully) WGCNA (weighted gene co-expression network analysis)

Line 28:  Similarly, define VIGS (virus-induced gene silencing)

Introduction:

Lines 46-49.  This is a run-on / awkward sentence, especially the last portion.

Lines 47-48: It should be “3’-phosphoadenosine 5’-phosphosulfate”, not what is written.

Line 51:  First, it was cloned from two plant species.  And, as written it seems like two “species” of flavonoids.  Rewrite to reflect both the citation and information.

Line 57-59:  It would be “… SOT genes...” (plural).  And, the last part should be followed up by what was learned.  As is, it feels like a “throw away sentence with no added information.”

Line 65:  “..bad external environment…” is too vague a statement.  Please add details to explain.

Lines 71-72:  This sentence on genetic engineering is too vague and with little to no added information and seems incomplete.  Either re-write to be more clear and add meaning, or perhaps delete.

Lines 74-76:  It is not clear to me if the GBSOT4 gene is first identified as being involved in FOV resistance in this manuscript OR in citation/reference #23, also from this research group,  Regarding this question and reference #23, I searched many different standard online directors/indexes and sources and even at the journal “molecular plant breeding” journal website, and I was not able to find this cited article.  Please double-check that the full reference information is correct.  Perhaps also add the “doi” website link.

Line 85, two issues:  First, as written this sentence is awkward and unclear regarding what was “preliminarily understood” by metabolomics.  Second, It should be, “This study provides new insights…”

Results:

Lines 90 – 92:  There needs to be a better introductory sentence to the results and this needs to include more information about the experiment, such as what plant species and specific SOT gene/protein used for the query / search to find the many cotton SOT genes/proteins.

Line 100:  replace “is” with “was likely”, for the data can suggest and are not so definitive to know with absolute certainty that gene loss is the explanation for the difference in gene number.

Line 105:  Enzymes with similar functions that have a such a huge range of amino acid number/length from “60” to 672 is fairly unusual. The original proteins from Arabidopsis also include a predicted SOT with only 60 amino acids, but it is not clear in Arabidopsis if the “60 aa” protein is functional or even expressed.  This issue was not mentioned or addressed in the manuscript and should be.  The fact that there are so many genes/proteins in cotton means it would not be surprising if some of these were cryptic and non-functional pseudo genes.

Figure 1, Line 123:  Of the ten figures, this is the only one that any of the text can be read at all.  However, the color indications of which genes correspond to which phylogenetic accession and species, based on color, is impossible to clearly see even in Figure 1.

Note, all of the rest of figures (2 – 10) the text is impossible to read, and thus the reader can not assess the data at all.  This major issue was mention above in the opening comments.

Line 141: As mentioned above, all numbers less than 10 should be written out in word form.  This would be the case for “nine” and “eight”.

Lines 162 – 171:   The authors should not somewhere, and make it clear in the writing, that the predicted cis-acting elements in the promoters using upstream activating promoter sequences are only speculative hypothesis, and none are known to be functional as predicted until tested experimentally. Further, gene expression regulation can also be controlled in other ways that via promoters / transcription, so absence of predicted cis-acting elements in promoter region does not rule other the RNA/protein can be controlled through other means (RNA stability, processing, protein regulation, etc…).  These issues should be mentioned either in the results or discussion section.

Line 174, Figure Legend 3:  In addition to not being able to read any of the text or data in this figure (as previously state), the legend needs more information to explain what is shown in this figure.

Line 180:  Define WGD on first use.

Line 195:  IT seems most likely the word is supposed to be “multicollinearity”.

Lines 215-216:  Write as “one”, not “1”.  And, it should then be “…this homologous gene pair, “ (singular).

Line 252:  Define “DPA”.

Line 273:  should be, “…SOT genes…” plural.

Lines 294 – 309:  Are the RNA-seq transcriptomic data being tested in this section still from reference #25?  If so, it would be good to re-state that.

Line 299: It should be, “heat stress”, not “hot stress.”

Line 310:  “fully utilized” for what?  Presumably as a food source, but this should be stated.

Line 312:  Cite source of data.

Line 318:  Defibne TDZ, presumably the thidiazuron herbicide.

Line 325:  Because these are all correlations and not absolute causation, the word “was” should be changed to “may be”.

Line 332, Figure Legend 6:  Define DPA.

Line 345:  The RNA expression and plant species are switched. It should read, “… level of GBSOT4 in Xinhai14…”, not the other way around.

Line 365:  Word order should be change to be more clear, to read, “Finally, we screened …”

Line 366:  Define WGCNA.

Line 385: Should it be “Amino sugar” or Amino acid sugars?

Line 406:  Define VIGS.

Line 419:  Figure 9c (again, impossible to read text) shows pictures of what I assume to be the “pTRV2-GBSOT4” and not also the comparison/controls to assess disease index values.  Can not see clearly enough in figures if this is true, and again, need to show controls clearly / magnified as well.

Line 427-428:  Change wording from use of “guessed”, and rest of this sentence is not clear at all.  Please rewrite to be more accurate and clear to readers.

Line 436:  First, write out in word form “six”.  And, next, fully explain what those six samples are.  As it, it is not clear at all to the reader what these samples are, and the impossible to read figure 10 complicates this.

Line 438: What is the “CK” control.  This is not explained anywhere else that I could find.

Line 441:  Define “OPLS-DA”

Line 443:  Delete “above” for it is not needed and, in fact, the data in Figure 10 are “below” this line.

Lines 450-457:  Are any of these metabolites sulfonated and have anything to do with the SOT genes/proteins?

Line 456:  italicize G. barbadense.

Line 470:  Define “DEM”

Discussion:

Line 521:  Based on the correlative and speculative nature of the data, change “showed” to “suggest” to be more accurate.

Line 525: Change “is” to “can be” to more accurately reflect that not all changes in expression patterns correlate with function.

Line 538:  Based on the correlative and speculative nature of the data, change “indicating” to “suggesting” to be more accurate.

Lines 586-602:  An interesting paragraph, but it is not clear that any of this is related to SOT genes/proteins.  Authors should make it more clear what link, if any, these have to the SOT’s, the focus of this paper.

Materials and Methods:

Line 612:  Looks like error in addition of “are analyzed” at end of this sentence.  If not the case, rewrite so it is clear.

Line 643/Section 4.5: Regarding the “Pheatmaps” in Figure 5, information about what software, likely “R”, was used to generate these.

Line 654:  Cite “published transcriptome data” source here.  Reference #25?

Line 678: In text authors used VIGS abbreviation a lot, but then in the methods section discussion this, it was not used at all.  Best to be consistent.

Line 682:  Italicize E. coli AND explain why the “connected product” (unusual way to say plasmid/construct) was sent for sequencing.  Presumably to confirm the correct construct with correct/exact sequence.

Line 684:  More explanation of the three constructs used for the VIGS experiments, including full explanation (or citation of reference if it came from other paper) for the pTRV2-CLA positive control.

Line 697:  Word “data” is plural, so it should read, “…data were…”

See above.

Author Response

Response to Reviewer 2 Comments

Dear Professor: Thank you very much for your comments on the manuscript. We have taken your suggestions in consideration point by point and made the updating of this revision.

Point 1: The authors have presented molecular genetic, transcriptomic, and metabolomic data about the predicted role of sulfotransferases, specifically GBSOT4 having a role resistance to critical cotton fungal pathogen Fusarium oxysporum. Overall the information and experimental design and process seems solid. However, there are some significant writing quality/clarity issues. And, most importantly, none of the text in the figures were readable due to small and very-low resolution.   Thus, it was impossible to fully review this manuscript for as a reviewer the provided data in the figures was not verifiable due to the text in the figures being impossible to read.   This meant the read could not know what genes / proteins / labels went with what data and information.   For this reason, there a significant redesign / presentation of all figures is needed before a complete review is possible.

Response 1: Thank you so much for your suggestion. As for the pictures that have been re-uploaded in the original text, I also uploaded the original pictures to the attachment during the previous submission process.

Point 2: Finally, in the section 2.13 (Lines 450-457) that address metabolomic changes related to FOV infection and possible resistance, it is not clear what, if any, of these have to do directly with SOT genes/proteins. Many of the metabolites mentioned are not related at all, at least none that are described here, to SOT function. The authors need to make a stronger an d more clear connection to how any of the metabolites with changes in abundance, including flavonoids, would relate to SOT genes/proteins and FOV resistance.Otherwise, it is correlation and not necessarily causation.   Similar issue for Lines 586-602 in the Discussion.

Response 2: Thank you so much for your suggestion. We added a clarification to this,We modified in the manuscript.121/5000

We have added details such as: RNA quality and integrity check, how is performed? A lot of details about cDNA synthesis, volume of RNA used

Point 3: Line10: Since “SOT” in this use is referring to proteins, it should NOT be italicized.   Later, when it refers to “SOT” genes, then it should (and usually correctly is) italicized.   Please check this issue throughout the manuscript, for often “SOT” is incorrectly italicized when also referring to proteins.

Response 3:Thank you so much for your suggestion. It has been amended in the original.

Point 4:Line 15:  “… for the first time…” as it is stated currently implies it is the first time to “use bioinformatic methods…”. Suggest rewrite to clarify.

Response 4:Thank you so much for your suggestion. It has been amended in the original.

Point 5: Line 21:  Depending on the journal, often “cis” is italicized, since it is a Latin term.  This is an issue throughout the manuscript.

Response 5:Thank you so much for your suggestion. It has been amended in the original.

Point 6:Line 22:  Should be “four”, as numbers less than 10 should be written out in word form, as was correctly done in Line 18.

Response 6:Thank you so much for your suggestion. It has been amended in the original.

Point 7: Line 26:  Define (write out fully) WGCNA (weighted gene co-expression network analysis)

Response 7:Thank you so much for your suggestion. It has been amended in the original.

Point 8:Line 28:  Similarly, define VIGS (virus-induced gene silencing)

Response 8:Thank you so much for your suggestion. It has been amended in the original.

Point 9: Lines 46-49.  This is a run-on / awkward sentence, especially the last portion.

Lines 47-48: It should be “3’-phosphoadenosine 5’-phosphosulfate”, not what is written.

Response 9:Thank you so much for your suggestion. It has been amended in the original.

Point 10:Line 51: First, it was cloned from two plant species. And, as written it seems like two “species”of flavonoids. Rewrite to reflect both the citation and information.

Response 10:Thank you so much for your suggestion. It has been amended in the original.

Point 11: Line 57-59: It would be “… SOT genes... ”(plural). And, the last part should be followed up by what was learned. As is, it feels like a“throw away sentence with no added information.”

Response 11:Thank you so much for your suggestion. It has been amended in the original.

Point 12: Line 65:  “.. bad external environment…” is too vague a statement.   Please add details to explain.

Response 12:Thank you so much for your suggestion. It has been amended in the original.

Point 13: Lines 71-72: This sentence on genetic engineering is too vague and with little to no added information and seems incomplete. Either re-write to be more clear and add meaning, or perhaps delete.

Response 13:Thank you so much for your suggestion. It has been amended in the original.

Point 14: Lines 74-76: It is not clear to me if the GBSOT4 gene is first identified as being involved in FOV resistance in this manuscript OR in citation/reference #23, also from this research group,  Regarding this question and reference #23, I searched many different standard online directors/indexes and sources and even at the journal “molecular plant breeding” journal website, and I was not able to find this cited article. Please double-check that the full reference information is correct. Perhaps also add the“doi”website link.

Response 14:Thank you so much for your suggestion. This is a Chinese journal, I have sent the "DOI" of the revised article to you, please check.DOI: 10.13271/j.mpb.020.001597.

Point 15: Line 85, two issues: First, as written this sentence is awkward and unclear regarding what was “preliminarily understood”by metabolomics. Second, It should be, “This study provides new insights…”.

Response 15:Thank you so much for your suggestion. It has been amended in the original.

Point 16: Lines 90 – 92:  There needs to be a better introductory sentence to the results and this needs to include more information about the experiment, such as what plant species and specific SOT gene/protein used for the query / search to find the many cotton SOT genes/proteins.

Response 16:Thank you so much for your suggestion. It has been amended in the original.

Point 17: Line 100:  replace “is” with “was likely”, for the data can suggest and are not so definitive to know with absolute certainty that gene loss is the explanation for the difference in gene number.

Response 17:Thank you so much for your suggestion. It has been amended in the original.

Point 18: Line 105:  Enzymes with similar functions that have a such a huge range of amino acid number/length from “60” to 672 is fairly unusual. The original proteins from Arabidopsis also include a predicted SOT with only 60 amino acids, but it is not clear in Arabidopsis if the “60 aa” protein is functional or even expressed.  This issue was not mentioned or addressed in the manuscript and should be.  The fact that there are so many genes/proteins in cotton means it would not be surprising if some of these were cryptic and non-functional pseudo genes.

Response 18:Thank you so much for your suggestion.To be honest, I don't understand what content you want me to modify here. I think what I wrote is OK. In a similar study in Ref. 11, he did the SOT gene of upland cotton. In this case, I'm making sea island cotton, and I'm making something similar to him.

Point 19: Figure 1, Line 123:  Of the ten figures, this is the only one that any of the text can be read at all.  However, the color indications of which genes correspond to which phylogenetic accession and species, based on color, is impossible to clearly see even in Figure 1.

Note, all of the rest of figures (2 – 10) the text is impossible to read, and thus the reader can not assess the data at all.  This major issue was mention above in the opening comments.

Response 19:Thank you so much for your suggestion. It has been amended in the original.

Point 20: Line 141: As mentioned above, all numbers less than 10 should be written out in word form.  This would be the case for “nine” and “eight”.

Response 20:Thank you so much for your suggestion. It has been amended in the original.

Point 21: Lines 162 – 171:   The authors should not somewhere, and make it clear in the writing, that the predicted cis-acting elements in the promoters using upstream activating promoter sequences are only speculative hypothesis, and none are known to be functional as predicted until tested experimentally. Further, gene expression regulation can also be controlled in other ways that via promoters / transcription, so absence of predicted cis-acting elements in promoter region does not rule other the RNA/protein can be controlled through other means (RNA stability, processing, protein regulation, etc…).  These issues should be mentioned either in the results or discussion section.

Response 21:Thank you so much for your suggestion. It has been amended in the original.

Point 22: Line 174, Figure Legend 3:  In addition to not being able to read any of the text or data in this figure (as previously state), the legend needs more information to explain what is shown in this figure.

Response 22:Thank you so much for your suggestion. It has been amended in the original.

Point 23: Line 180:  Define WGD on first use.

Response 23:Thank you so much for your suggestion. It has been amended in the original.

Point 24: Line 195:  IT seems most likely the word is supposed to be “multicollinearity”.

Response 24:Thank you so much for your suggestion. It has been amended in the original.

Point 25: Lines 215-216:  Write as “one”, not “1”.  And, it should then be “…this homologous gene pair, “ (singular).

Response 25:Thank you so much for your suggestion. It has been amended in the original.

Point 26: Line 252:  Define “DPA”.

Response 26:Thank you so much for your suggestion. It has been amended in the original.

Point 27: Line 273:  should be, “…SOT genes…” plural.

Response 27:Thank you so much for your suggestion. It has been amended in the original.

Point 28: Lines 294 – 309:  Are the RNA-seq transcriptomic data being tested in this section still from reference #25?  If so, it would be good to re-state that.

Response 28:Thank you so much for your suggestion. It has been amended in the original.

Point 29: Line 299: It should be, “heat stress”, not “hot stress.”

Response 29:Thank you so much for your suggestion. It has been amended in the original.

Point 30: Line 310:  “fully utilized” for what?  Presumably as a food source, but this should be stated.

Response 30:Thank you so much for your suggestion. It has been amended in the original.

Point 31: Line 312:  Cite source of data.

Response 31:Thank you so much for your suggestion. It has been amended in the original.

Point 32: Line 318:  Defibne TDZ, presumably the thidiazuron herbicide.

Response 32:Thank you so much for your suggestion. It has been amended in the original.

Point 33: Line 325:  Because these are all correlations and not absolute causation, the word “was” should be changed to “may be”.

Response 33:Thank you so much for your suggestion. It has been amended in the original.

Point 34: Line 332, Figure Legend 6:  Define DPA.

Response 34:Thank you so much for your suggestion. It has been amended in the original.

Point 35: Line 345:  The RNA expression and plant species are switched. It should read, “… level of GBSOT4 in Xinhai14…”, not the other way around.

Response 35:Thank you so much for your suggestion. It has been amended in the original.

Point 36: Line 365:  Word order should be change to be more clear, to read, “Finally, we screened …”

Response 36:Thank you so much for your suggestion. It has been amended in the original.

Point 37: Line 366:  Define WGCNA.

Response 37:Thank you so much for your suggestion. It has been amended in the original.

Point 38: Line 385: Should it be “Amino sugar” or Amino acid sugars?

Response 38:Thank you so much for your suggestion. it be “Amino sugar

Point 39: Line 406:  Define VIGS.

Response 39:Thank you so much for your suggestion. It has been amended in the original.

Point 40: Line 419:  Figure 9c (again, impossible to read text) shows pictures of what I assume to be the “pTRV2-GBSOT4” and not also the comparison/controls to assess disease index values.  Can not see clearly enough in figures if this is true, and again, need to show controls clearly / magnified as well.

Response 40:Thank you so much for your suggestion. It has been amended in the original.

Point 41: Line 427-428:  Change wording from use of “guessed”, and rest of this sentence is not clear at all.  Please rewrite to be more accurate and clear to readers.

Response 41:Thank you so much for your suggestion. It has been amended in the original.

Point 42: Line 436:  First, write out in word form “six”.  And, next, fully explain what those six samples are.  As it, it is not clear at all to the reader what these samples are, and the impossible to read figure 10 complicates this.

Response 42:Thank you so much for your suggestion. It has been amended in the original.

Point 43: Line 438: What is the “CK” control.  This is not explained anywhere else that I could find.

Response 43:Thank you so much for your suggestion. It has been amended in the original.

Point 44: Line 441:  Define “OPLS-DA”

Response 44:Thank you so much for your suggestion. It has been amended in the original.

Point 45: Line 443:  Delete “above” for it is not needed and, in fact, the data in Figure 10 are “below” this line.

Response 45:Thank you so much for your suggestion. It has been amended in the original.

Point 46: Lines 450-457:  Are any of these metabolites sulfonated and have anything to do with the SOT genes/proteins?

Response 46:Thank you so much for your suggestion. It should be related, but there is no further experiment to verify, this is our future work.

Point 47: Line 456:  italicize G. barbadense.

Response 47:Thank you so much for your suggestion. It has been amended in the original.

Point 48: Line 470:  Define “DEM” 

Response 48:Thank you so much for your suggestion. It has been amended in the original.

Point 49: Line 521:  Based on the correlative and speculative nature of the data, change “showed” to “suggest” to be more accurate.

Response 49:Thank you so much for your suggestion. It has been amended in the original.

Point 50: Line 525: Change “is” to “can be” to more accurately reflect that not all changes in expression patterns correlate with function.

Response 50:Thank you so much for your suggestion. It has been amended in the original.

Point 51: Line 538:  Based on the correlative and speculative nature of the data, change “indicating” to “suggesting” to be more accurate.

Response 51:Thank you so much for your suggestion. It has been amended in the original.

Point 52: Lines 586-602:  An interesting paragraph, but it is not clear that any of this is related to SOT genes/proteins.  Authors should make it more clear what link, if any, these have to the SOT’s, the focus of this paper.

Response 52:I'm sorry, I don't know what you want me to add, I think I have written clearly, because in other people's research also wrote this way.

In this study, sea island cotton was used as experimental material, normal plants were used as control, and plants with silted SOT gene were used as experimental samples. We conducted metabolomic analysis of both, because SOT gene is a member of the flavonoid pathway and plays a very important role in the process of plant stress resistance. The purpose of doing so is to observe which metabolites are related to the disease resistance of island cotton, and the specific relationship needs to be further verified by experiments.

Point 53: Line 612:  Looks like error in addition of “are analyzed” at end of this sentence.  If not the case, rewrite so it is clear.

Response 53:Thank you so much for your suggestion. It has been amended in the original.

Point 54: Line 643/Section 4.5: Regarding the “Pheatmaps” in Figure 5, information about what software, likely “R”, was used to generate these.

Response 54:Thank you so much for your suggestion. It has been amended in the original.

Point 55: Line 654:  Cite “published transcriptome data” source here.  Reference #25?

Response 55:Thank you so much for your suggestion. It has been amended in the original.

Point 56: Line 678: In text authors used VIGS abbreviation a lot, but then in the methods section discussion this, it was not used at all.  Best to be consistent.

Response 57:Thank you so much for your suggestion. It has been amended in the original.

Point 57: Line 682:  Italicize E. coli AND explain why the “connected product” (unusual way to say plasmid/construct) was sent for sequencing.  Presumably to confirm the correct construct with correct/exact sequence.

Response 57:Thank you so much for your suggestion. It has been amended in the original.

Point 58: Line 684:  More explanation of the three constructs used for the VIGS experiments, including full explanation (or citation of reference if it came from other paper) for the pTRV2-CLA positive control.

Response 58:Thank you so much for your suggestion. It has been amended in the original.

Point 59: Line 697:  Word “data” is plural, so it should read, “…data were…”

Response 59:Thank you so much for your suggestion. It has been amended in the original.

Thanks a lot for your patience and consideration.

Sincerely yours,

Jiao Yang

Postal address: College of Agriculture, Xinjiang Agricultural University, 311 Nongda East Road, Urumqi, 830052, China.

Telephone number: 18599052306

Fax number: 0991-8762263

E-mail address:[email protected]

Reviewer 3 Report

In this study, Authors investigated the role of a specific gene, GBSOT4, in enhancing the resistance of Gossypium barbadense (a cotton species) to Fusarium oxysporum f. sp. vasinfectum (FOV), a pathogenic fungus. 

The authors have done a wonderful job by highlighting the role of GBSOT4 gene by

Identification of SOT Genes,

Phylogenetic Analysis,

Gene Mapping and Visualization,

Transcriptome Analysis,

qRT-PCR Analysis,

Therefore the study suggests that GBSOT4 plays a crucial role in enhancing the resistance of G. barbadense to FOV, possibly through the regulation of flavonoid accumulation. Silencing this gene resulted in increased susceptibility to the pathogen.

The only comment i have is that the Figure resolution is very low, so high quality figures must be submitted.

Author Response

Response to Reviewer 3 Comments

Dear Professor: Thank you very much for your comments on the manuscript. We have taken your suggestions in consideration point by point and made the updating of this revision.

Point 1: In this study, Authors investigated the role of a specific gene, GBSOT4, in enhancing the resistance of Gossypium barbadense (a cotton species) to Fusarium oxysporum f. sp. vasinfectum (FOV), a pathogenic fungus.

The authors have done a wonderful job by highlighting the role of GBSOT4 gene by

Identification of SOT Genes,

Phylogenetic Analysis,

Gene Mapping and Visualization,

Transcriptome Analysis,

qRT-PCR Analysis,

Therefore the study suggests that GBSOT4 plays a crucial role in enhancing the resistance of G. barbadense to FOV, possibly through the regulation of flavonoid accumulation. Silencing this gene resulted in increased susceptibility to the pathogen.

The only comment i have is that the Figure resolution is very low, so high quality figures must be submitted.

Response 1: Thank you so much for your suggestion. We've reuploaded the picture.

Thanks a lot for your patience and consideration.

Sincerely yours,

Jiao Yang

Postal address: College of Agriculture, Xinjiang Agricultural University, 311 Nongda East Road, Urumqi, 830052, China.

Telephone number: 18599052306

Fax number: 0991-8762263

E-mail address:[email protected]

Round 2

Reviewer 2 Report

Plants Manuscript #:  2595713

Authors: Z. Su et al., 2023

Title:  GBSOT4 Enhances the Resistance of Gossypium barbadense to Fusarium ozysporum F. sp. vasinfectum (FOV) by Regulating the Content of Flavanoid

This is a re-review of the manuscript.  The authors have addressed the majority of my comments about the writing and written interpretation of the data/results.  However, one of the biggest issues I raised was inability to be able to read the text / font in the figures  due to small and (for some) low resolution.  Some of the re-uploading of figures helped, but for many (Figures 2, 4, 5, 6, 8 and 10) this is still an issue.  To be clear, I magnified the PDF with the images to the maximum and even then I could not read these.  If this were published and in normal print/view these figures would be impossible for readers to be able to read.  Thus, as I see it, these still need to be redone/reworked before this work could be published.

            In addition, I had asked that the authors provide some argument/reasoning for why the information provided in section 2.13 (Lines 457-465 in new version and Lines 450-457 in old version) that addresses metabolomic changes related to FOV infection and possible resistance, is related to with SOT genes/proteins and sulfonation that is the focus of this paper.  Many of the metabolites mentioned are not related at all, at least none that are described here, to SOT function.  As I see it, the authors still need to make a stronger and more clear connection to how any of the metabolites with changes in abundance, including flavonoids, relate to SOT genes/proteins and FOV resistance.  Otherwise, it is correlation and not necessarily causation.  Similar issue for Lines 602-611 in new version’s Discussion (Lines 586-602 in old version).

            Lastly, in my first set of comments I had asked the authors to include a magnified picture of the control samples leaves in Figure 9c, to be able to make a direct comparison to the magnified picture of the VIGS treated plants.  Considering the small size of the main pictures, I still feel this is needed.

There are still a few detailed writing issues that need to be changed:

Line 60:  need to add a “space” between “studied, such as…”.  There are few more locations in the manuscript that spaces are needed, particularly in the newly added text.

Line 177:  cis” needs to be italicized in figure legend.

Line 442:  Should be “six” instead of “6”.

Line 653:  Data is plural, so should read, “Using these data …”

These are included in the above comments.

Author Response

Response to Reviewer 2 Comments

Title:  GBSOT4 Enhances the Resistance of Gossypium barbadense to Fusarium ozysporum F. sp. vasinfectum (FOV) by Regulating the Content of Flavanoid

1.In addition, I had asked that the authors provide some argument/reasoning for why the information provided in section 2.13 (Lines 457-465 in new version and Lines 450-457 in old version) that addresses metabolomic changes related to FOV infection and possible resistance, is related to with SOT genes/proteins and sulfonation that is the focus of this paper.  Many of the metabolites mentioned are not related at all, at least none that are described here, to SOT function.  As I see it, the authors still need to make a stronger and more clear connection to how any of the metabolites with changes in abundance, including flavonoids, relate to SOT genes/proteins and FOV resistance.  Otherwise, it is correlation and not necessarily causation.  Similar issue for Lines 602-611 in new version’s Discussion (Lines 586-602 in old version).

Response 1:Thank you so much for your suggestion.

This SOT gene is a member of the flavonoid pathway, which plays an important role in plant resistance and has been reported in the previous literature.

This study is based on VIGS technology, and after silencing SOT gene, it does have some effects on cotton disease resistance. The figure also shows photos of phenotypes. So we sent metabolomics, which measures the metabolomics of flavonoids, and found that some metabolites that might be related to flavonoids were elevated or decreased. We looked at the KEGG metabolic pathway map, and did not find a direct relationship between SOT gene and the metabolites mentioned in the article, but it is not necessarily no indirect relationship. We can not provide direct evidence of the relationship between SOT gene and these metabolites, but we think that there is a certain relationship, otherwise why after the silence of SOT gene, The phenotype of disease resistance is obvious, and the related metabolite content will change?

As for the direct relationship, it is the subject of subsequent research.

2.This is a re-review of the manuscript.  The authors have addressed the majority of my comments about the writing and written interpretation of the data/results.  However, one of the biggest issues I raised was inability to be able to read the text / font in the figures  due to small and (for some) low resolution.  Some of the re-uploading of figures helped, but for many (Figures 2, 4, 5, 6, 8 and 10) this is still an issue.  To be clear, I magnified the PDF with the images to the maximum and even then I could not read these.  If this were published and in normal print/view these figures would be impossible for readers to be able to read.  Thus, as I see it, these still need to be redone/reworked before this work could be published.

Response 2:Thank you so much for your suggestion.

In my case, I think the picture is very clear, I have zoomed in, it is very clear.

I have sent the TIFF format of the original image to the editor, and he will send it to you.

3.Lastly, in my first set of comments I had asked the authors to include a magnified picture of the control samples leaves in Figure 9c, to be able to make a direct comparison to the magnified picture of the VIGS treated plants.  Considering the small size of the main pictures, I still feel this is needed.

Response 3:Thank you so much for your suggestion.

I'm sorry that this picture is not clearer, which brings obstacles to your review.

4.Line 60:  need to add a “space” between “studied, such as…”.  There are few more locations in the manuscript that spaces are needed, particularly in the newly added text.

Line 177:  “cis” needs to be italicized in figure legend.

Line 442:  Should be “six” instead of “6”.

Line 653:  Data is plural, so should read, “Using these data …”

Response 4:Thank you so much for your suggestion.

The above questions have been revised and marked in red in the original text.

Thanks a lot for your patience and consideration.

Sincerely yours,

Jiao Yang

Postal address: College of Agriculture, Xinjiang Agricultural University, 311 Nongda East Road, Urumqi, 830052, China.

Telephone number: 18599052306

Fax number: 0991-8762263

E-mail address:[email protected]